# Histone Deacetylase Functions in Gastric Cancer: Therapeutic Target?

**DOI:** 10.3390/cancers14215472

**Published:** 2022-11-07

**Authors:** Amandine Badie, Christian Gaiddon, Georg Mellitzer

**Affiliations:** Laboratoire Streinth, Université de Strasbourg, Inserm UMR_S 1113 IRFAC, 67200 Strasbourg, France

**Keywords:** histone deacetylase, histone deacetylase inhibitor, gastric cancer, biomarker

## Abstract

**Simple Summary:**

Our knowledge about the identity of many cancers has increased greatly during the last years and progress in their early identification as well as treatment options led to a net increase in the survival of many cancer patients. Unfortunately, gastric cancer does not belong to these cancers as it is still very badly treated and the chances to survive it are very low, less than 25%. This is mainly due to the fact that currently there are no possibilities to detect it at early stages and that tumors of gastric cancer patients seem all to be more or less different. In this respect, our knowledge about the differences between the gastric cancer from one patient to another is very limited. However, one family of proteins called “Histone Deacetylases” or HDACs, in contrast, seem to be present or their function altered in gastric cancers. This review summarizes our current knowledge about their role in gastric cancer development and their potential as an early detection marker and target to develop new treatment options.

**Abstract:**

Gastric cancer (GC) is one of the most aggressive cancers. Therapeutic treatments are based on surgery combined with chemotherapy using a combination of platinum-based agents. However, at metastatic stages of the disease, survival is extremely low due to late diagnosis and resistance mechanisms to chemotherapies. The development of new classifications has not yet identified new prognostic markers for clinical use. The studies of epigenetic processes highlighted the implication of histone acetylation status, regulated by histone acetyltransferases (HATs) and by histone deacetylases (HDACs), in cancer development. In this way, inhibitors of HDACs (HDACis) have been developed and some of them have already been clinically approved to treat T-cell lymphoma and multiple myeloma. In this review, we summarize the regulations and functions of eighteen HDACs in GC, describing their known targets, involved cellular processes, associated clinicopathological features, and impact on survival of patients. Additionally, we resume the in vitro, pre-clinical, and clinical trials of four HDACis approved by Food and Drug Administration (FDA) in cancers in the context of GC.

## 1. Introduction

Gastric cancer (GC) is a public health problem. It represents the fourth leading cause of cancer-related death in the world, after lung, colon, and liver cancers, with around 769,000 deaths in 2020 (7.7% of all cancers) [1]. In terms of incidence, GC is the fifth most common cancer with more than one million new cases diagnosed in 2020 (5.6% of all cancers) [1]. The incidence and mortality of GC vary by region with an especially high incidence rate in Eastern Asia and in Eastern Europe.

The treatment of GC depends primarily on the stage of the disease [2]. At early stages (Stage II or less) patients undergo resection procedures to remove the malignancy. Depending on the location of the tumor and the depth of the invasion, endoscopic mucosal resection, distal esophagectomy, subtotal, or total gastrectomy are considered. For locally advanced diseases (clinically T2–4 or positive lymph node), surgery is combined with preoperative chemotherapy or perioperative chemoradiotherapy using a combination of platinum-based agent, usually oxaliplatin, and a cytotoxic compound such as 5-fluorouracil (5-FU). At metastatic stages, first-line chemotherapies are applied and are combined with targeted therapy against HER2 or immunotherapies against PD-L1/PD-1 immune checkpoint depending on their expression level. Unfortunately, the five-year survival rate for the GC patients is less than 30% due to the high inter- and intratumor heterogeneity [3], and most diagnoses occur during late stages of the disease and are met with high rates of chemotherapeutic resistance [4]. It is therefore evident that there is still much effort needed to identify new prognostic factors and therapeutic options to increase the survival rate for GC patients.

GC is a multifactorial disease. There are infectious causes such as *Helicobacter pylori* (*H. pylori*) bacterium infection, which is the main risk of GC, and Epstein–Barr virus (EBV) infection, and environmental risks such as a high consumption of tobacco and alcohol or special diets [3]. It is well documented that a salty or smoked food diet and low consumption of fruits and vegetables increases the risk of developing GC [3]. Although rare, representing only 1 to 3% of GC cases, there are also genetic factors comprising three main syndromes: hereditary diffuse GC (HDGC), characterized by autosomal dominant transmission of *CDH1* mutation; gastric adenocarcinoma and proximal polyposis of the stomach (GAPPS), characterized by autosomal dominant transmission of fundic gland polyposis and mutations of tumor suppressor *APC*; and familial intestinal GC (FIGC), characterized by autosomal dominant inheritance pattern and intestinal-type GC [5,6]. Moreover, the risk of developing GC increases with age and is more important for men than women, with a 2:1 ratio [1].

Over the past years, different classifications of GC have been established. GCs are in 95% of cases adenocarcinomas and are highly heterogeneous. According to the Laurén histological classification, there are two main histological types of GC: intestinal and diffuse types, which are moderately differentiated with glandular structures or poorly differentiated, respectively [7]. The intestinal type is the most common one, but the diffuse type is suggested to be associated with aggressive stages and the worst prognosis [7,8]. In addition to this histological classification, molecular classifications have been established by the Cancer Genome Atlas researcher network (TCGA) and the Asian Cancer Research Group (ACRG). In 2014, TCGA proposed a molecular classification of GC with four genomic subtypes: Epstein–Barr virus-infected tumors (EBV, 9%), microsatellite instable tumors (MSI, 22%), genomically stable tumors (GS, 20%), and chromosomally unstable tumors (CIN, 50%) [9]. Studies showed that EBV-positive and MSI-high GCs have better prognosis [10,11,12]. In 2015, the ACRG proposed four molecular subtypes: MSI tumors (23%), which have the best prognosis; microsatellite stable and epithelial-to-mesenchymal transition phenotype tumors (MSS/EMT, 15%), which have the worst prognosis; and microsatellite stable and TP53-active tumors (MSS/TP53+, 26%) or no TP53 signature tumors (MSS/TP53-, 36%) [13]. However, although these molecular classifications clearly highlight the genetic heterogeneity of GCs, they are not used diagnostically in clinical practice and, so far, they have not led to the development of prognostic markers or new therapies.

In addition to these genetic alterations, increasing evidence in the literature points to an important implication of epigenetic processes in the development of GC. Among the epigenetic processes most studied and known to be involved in the development of cancers are DNA methylation and post-translational modifications of histones [14]. Among histone modifications, there are acetylation by histone acetyltransferases (HATs) and deacetylation by histone deacetylases (HDACs) [15]. HDAC alterations in various cancers lead to the development of HDAC inhibitors (HDACis) [16]. Several HDACis have already been clinically approved for the treatment of cancers such as T-cell lymphomas and multiple myeloma [17].

In this respect, the objective of this review is to give an overview on what is currently known about the expression and function of HDACs in GC.

## 2. Histone Deacetylases

### 2.1. Generalities

Acetylation and deacetylation of histones are epigenetic processes involved in the regulation of gene expression by acting on the chromatin conformation [14]. Histone acetylation is carried out by HATs, which catalyze the transfer of an acetyl group to the lysine residue of the histone amino terminal tail. This leads to the de-condensation of chromatin and allows the transcription of genes. On the contrary, the deacetylation of histones is carried out by HDACs, which catalyze the loss of this acetyl group on the amino terminal tail of histones, leading to the condensation of chromatin and the repression of gene transcription.

HDACs are divided into four classes based on their sequence homology with yeast HDACs (Table 1) [14,18]. Class I, homologous to the yeast Rpd3 protein, groups together HDAC1, 2, 3, and 8 showing a clear nuclear localization. Class II, homologous to the yeast Hda1 protein, is divided into two subclasses: subclass IIa grouping together HDAC4, 5, 7, and 9 and subclass IIb with HDAC6 and 10. In contrast to class I, HDACs of class II can have a nuclear or cytoplasmic localization. Class III, homologous to the yeast Sir2 protein, groups together all sirtuins (SIRTs), from 1 to 7, showing a nuclear, cytoplasmic, or mitochondrial localization. Finally, class IV has a single member, HDAC11, which has characteristics of both class I and class II. These four classes are divided into two families: classes I, II and IV, whose activity is dependent on zinc (Zn^2+^-dependent), and class III, whose activity is dependent on nicotinamide adenine dinucleotide (NAD^+^-dependent).

Importantly, in addition to their histone deacetylation activity, they also interact with and deacetylate non-histone proteins, such as p53, HIF-1α, STAT3, c-Myc, NF-κB, or estrogen receptors, altering directly their activity [19]. In addition, HDACs can interact with various cellular proteins, such as HSP90, preventing interaction of this later with the ligand-inducible transcription factor glucocorticoid receptor (GR) and inhibiting its transcriptional activity [19]. Through their varied targets, HDACs are involved in different biological processes such as cell cycle progression, proliferation, differentiation, and many more, and their alterations are associated with the development of many cancers [20].

### 2.2. In Gastric Cancer (GC)

#### 2.2.1. HDACs of Class I

**HDAC1:** Among class I HDACs, HDAC1 is the most studied and the majority of studies showed that HDAC1 is overexpressed in GC tissue [21,22,23,24,25,26,27,28,29,30]. An exception to this is a study performed by Wisnieski and co-workers on 50 paired (tumoral and non-tumoral) tissue samples of GC patients concluding that HDAC1 is less expressed in GC tissue compared to adjacent non-tumoral gastric tissue [31]. Unfortunately, when regarding the association of HDAC1 expression with clinicopathological parameters, not all results point in the same direction. According to some studies, high expression of HDAC1 in GC can be associated with age, Lauren’s classification, *H. pylori* infection, tumor size, lymphovascular invasion, lymph node metastasis (LNM), or even advanced tumor stages [24,25,26,28,29,32,33]. In addition, patients with a strong expression of HDAC1 seem to have a poorer overall survival and disease-free survival (DFS) than patients with low expression [26,28,29] even if this worse prognosis is not always statistically significant [33]. Indeed, Mutze and co-workers showed that a high HDAC1 expression is significantly correlated with a low overall survival only in GC patients responding to neoadjuvant chemotherapy (platinum/5-FU) [33]. On the contrary, the five-year survival does not change according to HDAC1 expression in the study of Eto and co-workers [23]. Based on in vitro experiments, HDAC1 is involved in the progression of GC using different signaling pathways (Figure 1). By its recruitment on different promoters, it can have a repressive activity on them. For example, HDAC1 promotes cell proliferation via the HDAC1/MORC2/p21 pathway [34]. However, *H. pylori* infection seems to reverse the repressive activity of HDAC1 on p21 promoter in GC cells and thus promotes acetylation of the latter, leading to p21 expression [35]. These results agree with observations made by Pero et al., which showed that *H. pylori* decreases expression and activity of several HDACs such as HDAC1 [36]. Moreover, other studies showed that HDAC1 diminishes caspase-2-dependent apoptosis via its interaction with CRADD (CASP2 and receptor-interacting protein kinase 1 domain containing adaptor with death domain) promoter [37]. Similarly, Regel and co-workers showed that by forming a complex with HDAC2, HDAC1 represses the expression of CITED2 (Cbp/p300-interacting transactivator with Glu/Asp-rich carboxy-terminal domain 2), which normally inhibits the transactivation activity of the coactivator protein 300 (EP300) and transcription of HIF1α-dependent resistance genes [38]. However, Jiang et al. showed that HDAC1 increases the activity of HIF-1α, a transcription factor involved in glycolysis, and therefore promotes glycolysis in GC [26]. In addition, the role of HDAC1 in GC also involves the regulation of different RNA families: long non-coding RNAs (lncRNAs) and micro-RNAs (miRNAs). On one hand, HDAC1 is known to upregulate the expression of lncRNAs BC01600 and AF116637, involved in cell proliferation, but on the other hand also to suppress the transcription of lncRNA similar to HRCEG, which suppresses cell proliferation and the EMT process promoting apoptosis [30,39]. About miRNA regulation, downregulation of HDAC1 has been shown to increase the expression of miR-34a involved in CD44 repression to promote cell adhesion, migration, and resistance to apoptosis [40]. Another miRNA is known to be involved in the regulation of HDAC1: miR-520h [41]. It is induced by doxorubicin and inhibits HDAC1 expression, enhancing the interaction of doxorubicin to DNA and consequently cell death. Thus, HDAC1 is a doxorubicin resistance factor in GC. In conclusion, HDAC1 seems to have a pro-tumoral role and is a drug resistance factor in GC despite the controversy about its link to patient’s survival.

**HDAC2:** Although not studied as much as HDAC1, expression data on different cohorts consistently showed that HDAC2 has an increased expression in GC [29,31,38,42,43,44,45,46], especially at advanced and metastatic stages of the disease according to Song et al [44]. However, in contrast to these publications, Nakagawa et al. found no difference in HDAC2 expression between GC and normal tissue [47]. Unfortunately, no clear consensus can be found concerning its expression according to the histological GC type. Mutze and co-workers found that its expression is associated with poor tumor differentiation and the non-intestinal type [33]. However, Song et al., Weichert et al., and Regel et al. found no difference in its expression between the intestinal and the diffuse type of GC [38,44,48]. Similarly, association of HDAC2 expression with overall survival seems, with the current available data, not to be straight forward. Mutze et al. showed that HDAC2 expression is not associated to overall survival except when homing in on the non-responders, where its high expression is associated with better survival [33]. In contrast to this, in a retrospective study performed by Weichert et al., three-year survival of GC patients dropped from 50% in the HDAC2 negative GC samples to 16% in samples expressing a high level of HDAC2 [48]. This is supported by Sun et al. showing a lower overall survival in GC patients with high HDAC2 expression [29]. In vitro, HDAC2 seems to be strongly expressed in many GC cell lines (ex. AGS, MKN-1, GES-1, BGC-823, NCI-N87), and inhibition of its expression or activity reduces cell proliferation and induces apoptotic and autophagic cell deaths [29,42,45,46]. Interestingly, HDAC2 can bind to the promoter of the cyclin-dependent kinase inhibitor p16^INK4a^ [42] or complexed with HDAC1 to the promoter of CITED2, involved in HIF-1α-dependent transcription, thereby inhibiting their expression [38]. Taken together, HDAC2 seems to be strongly expressed in GC at advanced stages, but its correlation with overall survival is not clear.

**HDAC3:** Compared to HDAC1 or 2, even far less is known about the expression of the third member of the class I HDACs namely HDAC3 and its role in tumor progression and aggressiveness in GC. In four Asian cohorts, HDAC3 expression was found to be higher in GC tissue compared to the matched non-tumoral tissue [22,49,50,51]. However, in one Asian and one Brazilian cohort, investigators found no difference in the expression level between tumoral and non-tumoral gastric tissue [31,47]. Knockdown of HDAC3 reduces cell growth, migration, invasion, and cell viability, and increases apoptosis in GC lines and reduces tumor growth in xenograft models using GC cell lines [49,50,51]. Three different molecular mechanisms are proposed. The first suggests that high overexpression of HDAC3 in GC cell lines induces the expression of miR-454, which targets the chromodomain helicase DNA binding protein 5 (CHD5), a reported tumor suppressor gene in various types of cancers including GC [49]. Importantly, high expression of HDAC3 correlates with a high expression of miR-454 whose expression inversely correlates with CHD5 expression. In addition, high miR-454 expression in GC patients correlates with an advanced tumor state and worse survival [49]. The second mechanism involves the FOXA2/FTO/MYC signaling axis, where HDAC3 inhibits FOXA2 activity thereby releasing its repressive effect on FTO and MYC expression [50]. Importantly, high HDAC3 expression in GC tissue correlates with high FTO and MYC expression, an advanced tumor state and lower survival probability [50]. A third mechanism suggests that HDAC3 negatively regulates miR-376-3p, which inhibits wingless-type MMTV integration site family member 2b (WNT2b) [51]. Additionally, high expression of HDAC3 correlates with a low expression of miR-376-3p and high expression of WNT2b, and strong HDAC3 expression is associated with tumor grade, tumor infiltration depth, LNM, and tumor stage in GC [51]. Wu et al. highlighted that a high level of HDAC3 is associated with tumor recurrence and poor prognosis [52]. Thus, a set of four publications present a pro-tumor function of HDAC3 with worse prognosis, opposing one study where no variation in its expression was observed.

**HDAC8:** So far, almost nothing is known about the last member of class I HDACs, HDAC8. There is one initial study showing no difference in the expression of HDAC8 between GC tissue and corresponding non-cancerous tissue [47]. In contrast to this, and this time not due to different ethnic origins of the cohorts, Song and co-workers found that HDAC8 is significantly upregulated in 92.2% of GC tissue compared to non-cancerous tissue [53]. High HDAC8 expression correlates with advanced tumor stages, LNM, and poor differentiation [53]. In the GC BGC823 cell line, inhibition of its expression reduces cell growth and colony formation and promotes cell cycle arrest in G0/G1 phase and the expression of the apoptotic markers, cleaved caspase-3 and 6 [53]. Conversely, overexpression of HDAC8 has the opposite effect. Taken together, HDAC8 seems to be higher expressed in GC correlating with advanced stages of the disease, but due to the sparsity of available data, no further conclusion can be drawn concerning its implication in GC progression and aggressiveness.

#### 2.2.2. HDACs of Class IIa

**HDAC4:** The good aspect about HDAC4 is that all expression studies agreed that HDAC4 displays an increased expression in gastric tumors [54,55,56,57,58]. Unfortunately, most studies were done only on small cohorts of patients. However, Spaety and co-workers have shown, by using the TCGA database, that HDAC4 expression varies depending on the molecular subgroup of GC (MSI, EBV, CIN, and GS) with a two-fold higher expression in the GS compared to the MSI subgroup [57]. Its expression tends also to be higher in the diffuse-type GC of Lauren’s classification, often associated with the GS subgroup [57]. Additionally, high HDAC4 expression is correlated with an advanced invasion of the gastric mucus, LNM, and TNM stage, and is associated with a shorter overall survival and DFS in GC patients [58]. Interestingly, HDAC4 seems to be a resistance factor in GC, because when it is overexpressed or inhibited by specific siRNAs, GC cells become more or less resistant to cisplatin, respectively [57]. Likewise, when Spaety and co-workers treated nude mice harboring intradermal tumors derived from HSC39 human GC cells with cisplatin and the HDAC4 inhibitor, LMK235, tumor growth was strongly decreased [57]. This is also in accordance with data published by Colarossi et al., who showed that HDAC4 inhibition increases the effect of docetaxel [54]. Importantly, and underlining the role of HDAC4 as resistance and pro-tumoral factor, is that when going deeper into the analysis of TCGA data on GC, Spaety and co-workers found that patients with a mutated *HDAC4* have an overall better survival than patients with a wild type *HDAC4* [57]. Mechanistically, silencing of HDAC4 inhibits cell proliferation, migration, and invasion via the HDAC4/ATG4B/MEKK3/p38 pathway [58] but favors cell cycle arrest and apoptosis, characterized by cleavage of caspase-3 and induction of proapoptotic genes, such as *BIK* or *p21*, in part via a p53-dependent mechanism [55,57]. In addition, lncRNAs and miRNAs have been shown to impact the HDAC4 expression in GC. Among these, there is the lncRNA MIAT that promotes cell proliferation, migration, and invasion via the MIAT/miR-29a-3p/HDAC4 pathway [56]. Finally, the tumor suppressor p53 is also able to promote HDAC4 expression by inhibiting miR-140 expression [57]. In conclusion, HDAC4 is a tumor promoter and chemoresistance factor in GC and alteration in HDAC4 status seems to be a predictive factor for the overall survival of GC patients.

**HDAC5:** Three published studies addressed the expression and function of HDAC5 in the tissue of GC patients. One of them showed that HDAC5 is less expressed in GC tissue compared to normal gastric tissue [43]. On the contrary, according to Zhang et al., HDAC5 is upregulated in 60 pairs of GC tissue [59]. Similarly, the third study shows that a high level of HDAC5 is associated with shorter overall survival of GC patients [60]. Additionally, Zhang and co-workers showed a proliferation and migration induction by HDAC5 partly regulated by HOXC-AS3, linked to Y-box-binding protein 1 (YBX1), at transcriptional level in GC cells [59]. These last data suggest that HDAC5 could be used as a therapeutic target and prognostic biomarker in human GC, as has been shown in human breast cancer [61].

**HDAC7:** Another less studied HDAC of the class IIa in GC is HDAC7. There are two papers describing its expression in GC tissue [62,63]. Unfortunately, the published data by Yu and co-workers lack some depth of clarity and their interpretation is delicate. However, according to Zhang et al., HDAC7 is more expressed in GC tissue compared to adjacent normal tissue, and its high expression is associated with a poor survival [63]. In addition, downregulation of HDAC7 in GC cell lines results in reduced cell viability and migration and invasion capacities [63]. Although these last data suggest that HDAC7 could play a pro-tumoral role in GC, further studies are needed to gain a clearer picture of its role in GC.

**HDAC9:** The good aspect about HDAC9 is that all three expression studies done so far, Xiong et al., 15 patients [64]; Xu et al., 63 patients [65]; and Wu et al., 80 patients [66], have concluded that HDAC9 is more expressed in GC tissue compared to non-tumoral tissue. Two of these studies also came to the conclusion that high HDAC9 expression correlates with a lower overall patient survival [64,65]. In contrast to this, Wu and co-workers suggested that high expression HDAC9 can neither be associated with a patient survival prognosis nor with any clinical parameters [66]. The latter contradicts Xu’s findings, which suggest that high HDAC9 expression correlates with advanced stages of GC [65]. Interestingly, HDAC9 expression level in the para-carcinoma tissue seems to correlate with some clinical parameters, as Wu and co-workers showed that it is negatively correlated with metastatic stages and positively with patient survival [66]. In vitro, transient inhibition of HDAC9 in GC cell lines reduces colony formation and proliferation and favors apoptosis induction [64,65]. Taken together, the potential pro-tumoral role of HDAC9 in GC still needs to be taken with care partly due to the contradictory clinical correlations and the small number of publications.

#### 2.2.3. HDACs of Class IIb

**HDAC6:** To our knowledge, out of the so far published studies, in which the role of HDAC6 in GC has been addressed, only two of them analyzed the expression of HDAC6 in human GC tissue [32,67]. Unfortunately, Park et al. and He et al. did not come to the same conclusion. Whereas Park et al. stated that HDAC6 is overexpressed in GC tissue [67], He et al. concluded that its expression is lower compared to adjacent normal tissue [32]. Both backed up their findings either by gene expression data analysis of the National Center for Biotechnology Information (NCBI) and Gene Expression Omnibus (GEO) databases [67], or by HDAC6 RT-qPCR done on 20 GC and 12 normal tissue samples [32]. In addition to these conflicting observations on the expression of HDAC6 in GC, they also came to different conclusions regarding its role in gastric carcinogenesis. Indeed, Park et al. showed that overexpression of HDAC6 promotes cell growth in GC [67]. This is supported by a positive correlation between HDAC6 and epidermal growth factor receptor (EGFR), whose signaling is involved in the regulation of cell growth and in the progression of many cancers [67]. HDAC6 has been shown to promote the activation of EGFR as well as to limit its degradation by inhibiting rabaptin-5 in GC [67]. Thus, these results suggest an oncogenic role for HDAC6 via the EGFR/rabaptin-5 pathway. On the contrary, according to He et al., high expression of HDAC6 is negatively associated with tumor progression and positively associated with overall survival in GC patients [32]. To explain this, the discrepancy might be due to the differences in the analyzed GC subtypes and/or stages, as He and co-workers showed that HDAC6 expression progressively reduces during progression from precancerous conditions to GC [32]. Further, the authors showed that HDAC6 is inversely correlated with *H. pylori* infection in gastric mucosal lesions [32]. In vitro and in vivo infection of GES-1 cell line or mice by SS1 strains decreases HDAC6 expression suggesting that *H. pylori* can be involved in its downregulation in GC [32]. In addition, some of the results should be taken with precaution as quantifications of Western blots were missing, sometimes only very small cohorts have been analyzed, and the methods used to confirm their HDAC6 protein expression findings were not the same. In conclusion, the expression of HDAC6 and its role in gastric carcinogenesis remains to be further investigated.

**HDAC10:** Similar to some of its other family members, HDAC10 expression and role in GC is not yet studied much. There is one study published in which the expression of HDAC10 was examined by immunohistochemistry on 179 paraffin-embedded GC tissue specimens [68]. Authors scored the immunohistochemical staining according to its intensity: 0 (negative), 1 (weakly positive), 2 (moderately positive), and 3 (strongly positive), and they concluded that out of the 179 analyzed samples, 87 were negative and 92 were positive (51.4%) for HDAC10 expression [68]. At the same time, out of 79 adjacent tissue samples analyzed, only 10 displayed no expression of HDAC10 whereas the other 69 adjacent tissue samples were positive (87.3%) [68]. Although interesting, due to the lack of additional studies, these data must be interpreted with some precaution as immunohistochemical staining are not always straight forward and may vary greatly according to the antibody and protocol used.

#### 2.2.4. HDACs of Class III

**SIRT1:** The regulation and role of SIRT1 in the framework of GC is complex and controversial in the literature. A set of studies showed that SIRT1 is upregulated in GC [69,70,71,72,73,74,75,76,77,78] while another set of studies concluded that its expression is diminished in GC [79,80,81,82]. In addition, its expression is correlated, either positively or negatively, with a set of clinicopathological parameters. More precisely, SIRT1 expression has been shown to be correlated with sex, age, histological type, grade, tumor size, tumor invasion, elevated serum levels of carcinoembryonic antigens, LNM, and TNM stage [69,70,71,72,73,74,77]. However, and as said, correlations made are not always the same. For example, its strong expression is associated with shorter overall and relapse-free survival and PFS [69,70,72,74,77]. However, three other studies stated that SIRT1 expression can be considered to be a good prognostic factor [71,79,83], especially when combined with the expression of deleted breast cancer gene 1 protein (DBC1) and β-catenin [71].

Based on in vitro and in vivo experiments, SIRT1 has been shown to promote cell proliferation, migration, EMT, autophagy, and apoptosis resistance [70,74,75,84,85]. This is partly enabled by miR-12129 and miR-204 inhibition in GC, which are involved in the post-transcriptional repression of SIRT1 [75,85]. In addition, SIRT1 expression seems to be positively associated with phospho-forkhead box protein O1 (pFOXO1), an inactive form of FOXO1 shown to inhibit growth and angiogenesis in GC via the HIF1α/VEGF pathway [86]. Thus, miR-204 and FOXO1 would act as negative regulators of SIRT1. However, a study by Yan et al. showed that SIRT1 expression in GC is preserved by yes-associated protein (Yap) and that the Yap/SIRT1/Mfn2 pathway promotes cell survival and migration via mitophagy [84]. Conversely, SIRT1 has also been shown to be involved in cell proliferation inhibition, G1 cell cycle arrest, migration and invasion, and apoptosis induction in vitro, and tumor growth and metastasis inhibition in vivo [78,80,82,87]. The anti-tumoral role of SIRT1 seems to involve NFκB/cyclin D1 [80], SIRT1/c-JUN/ARHGAP5 [87], or SIRT1/STAT3/MMP-13 pathways [78]. In addition, resveratrol, an activator of SIRT1, displays similar effects on cell viability and causes senescence via repression of STAT3 and NFκB activation by SIRT1-induced deacetylation [88,89]. As for the negative regulators of SIRT1, miR-543, -1301-3p, and -132 have been found to be upregulated in some GCs, and their overexpression leads to cell proliferation for miR-543 and -1301-3p and resistance to chemotherapy via an SIRT1/CREB/ABCG2 axis for miR-132 [81,90,91]. SIRT1 has also been shown to inhibit GC chemoresistance by activating the AMPK/FOXO3 pathway [79]. However, overexpression of ATF4, a stress response factor involved in cellular homeostasis, increases the expression of SIRT1 by binding to its promoter and resulting in an increased multi-drug resistance of GC cells [92]. Interestingly, Chen et al. showed that oxaliplatin reduces the deacetylation activity of SIRT1, thereby enhancing p53 acetylation and apoptosis [93].

Taken together, SIRT1 appears to play a dual role in gastric tumorigenesis. Under certain circumstances, it might have a pro-tumor function and might be used as an early diagnostic marker for a poor prognosis [69,70,72,74,77]. Whereas, in other cases, it could act as a tumor suppressor, being a good prognostic factor [71,78,79,80,82,83,87]. Likewise, in some cases drug resistance of GC cells seemed to depend on SIRT1 expression whereas in other cases it was the opposite. This controversial role might be explained by the number of samples included in the studies, the ethnic populations, the techniques used, the cellular context, the microenvironmental conditions, and many other criteria.

**SIRT2:** Not much is known about SIRT2 in GC. Only two studies addressed the function of SIRT2 in GC [94,95]. Li et al. showed that its expression is increased in GC tissue compared to adjacent normal tissue correlating with a reduced PFS and overall survival [94]. In vitro, knockdown of SIRT2 seems to decrease in part cell proliferation but significantly represses cell migration and invasion and induces apoptosis in GC cell lines [94,95]. In vivo, inhibition of SIRT2 suppresses tumor growth and the metastatic potential of a GC cell line in nude mice [94]. These functions of SIRT2 seem to be mediated in part by its deacetylation activity on phosphoenolpyruvate carboxykinase 1 (PEPCK1), leading to its stabilization and RAS/ERK/JNK/MMP-9 pathway activation [94]. To conclude, SIRT2 could have a pro-tumoral role, but more thorough investigation remains necessary.

**SIRT3:** The problem with SIRT3 is that there are conflicting published data. Four papers reported that SIRT3 expression is decreased in GC tissue compared to normal gastric tissue [96,97,98,99]. In contrast, the publication by Cui and co-workers showed that SIRT3 is more strongly expressed in GC tissue than in normal gastric tissue [100]. In part, this might be explained by the fact that only small cohorts of GC patients were analyzed in the four studies, making differences in age, sex, subtype, and treatment protocols between studies even more important. However, and adding to the discrepancy, Wang et al. and Ma et al. found that overexpression of SIRT3 in the GC AGS cell line reduces cell proliferation, colony formation capacity, and invasion, and decreases tumor growth in nude mice [97,98,101], whereas Cui and co-workers reported the opposite effect showing that overexpression of SIRT3 in the same GC cell line stimulates proliferation [100]. In terms of clinicopathological parameters, a meta-analysis performed by Yu and co-workers suggested that high SIRT3 expression correlates with a longer overall survival and low SIRT3 expression with a poor differentiation status [102]. Concerning the later, this might correlate with the finding of Huang et al., showing that decreased SIRT3 expression is associated with a lower overall survival, as poorly differentiated GC seems overall to be more aggressive than well-differentiated GC [96]. Molecularly, in GC cell lines, SIRT3 has been shown to repress Notch1, inhibiting its negative regulation of cell proliferation [97,98], to repress HIF1α [99,101], and to interact with lactate dehydrogenase A (LDHA), thereby increasing its activity and ROS homeostasis [100]. However, all this must be taken with precaution, due to the above-described discrepancies in the published data, the small size of the analyzed cohorts, and the small number of publications addressing the role of SIRT3 in GC.

**SIRT4:** As for SIRT2, not much is known about SIRT4 in GC. However, the two studies performed on human GC tissue agreed that its expression in GC tissue is significantly lower than in the adjacent normal tissue and that SIRT4 correlates with LNM and tumor stage [103,104]. In addition, Sun et al. showed that GC patients with low expression of SIRT4 have a poorer survival than patients with high expression [104]. In vitro, overexpression of SIRT4 inhibits cell proliferation, colony formation, migration, and invasion, whereas its inhibition has in part an opposite effect [104,105]. This effect seems to depend on E-cadherin as overexpression of SIRT4 stimulates its expression, whereas it is diminished in the absence of SIRT4, suggesting that SIRT4 might suppress EMT through regulation of E-cadherin [104]. Taken together, although not much studied, the current available data on the expression and function of SIRT4 in GC suggest that it might have tumor suppressor function in GC.

**SIRT5:** Similar to SIRT4, SIRT5 is, so far, not much studied in GC and available data suggest that it is lower expressed in GC tissue than adjacent normal tissue [106,107] correlating with lymphovascular invasion and a poorer overall survival [106]. In GC cell lines, overexpression of SIRT5 inhibits cell growth, arrests G1/S cell cycle transition, and suppresses migration and invasion [107,108]. According to two studies, this function seems to depend in part on SIRT5 desuccinylation activity on 2-oxoglutarate dehydrogenase (OGDH) and S100A10 [107,109]. To conclude, these published data suggest that similar to SIRT4, SIRT5 might have a tumor suppressor function in GC.

**SIRT6:** Unfortunately, only one published study addressed the expression and function of SIRT6 in GC. This study shows that, similar to SIRT4 and SIRT5, SIRT6 expression is lower in GC tissue when compared to normal gastric tissue, and that its decreased expression correlates with tumor grade, tumor size, and TNM stage, along with decreased overall survival and DFS of GC patients [110]. Likewise, overexpression of SIRT6 reduces cell proliferation and causes cell cycle arrest and apoptosis in vitro and reduces GC cell line tumor growth in nude mice [110]. In addition, Zhou and co-workers showed that its expression is inversely correlated with p-STAT3 expression in GC tissue and that it inhibits the JAK2/STAT3 pathway, known to positively regulate cell growth, in GC cells [110]. In short, similar to SIRT4 and SIRT5, SIRT6 might have a tumor suppressor function in GC.

**SIRT7:** Zhang and co-workers showed that SIRT7 is upregulated in 78% of GC tissue compared to adjacent gastric tissue [76]. Its expression is correlated with various clinicopathological factors such as the extent of the gastrectomy, tumor size, depth of tumor invasion, lymph nodes status, metastases, and tumor stage [76]. In addition, its high expression seems to be associated with low overall survival and DFS. Knockdown of SIRT7 in GC cells induces apoptosis and inhibits cell proliferation and colony formation in vitro and GC cell growth in vivo in nude mice [76]. These results suggest that SIRT7 favors gastric tumor progression, which is supported by Zhang et al.’s findings that SIRT7 binds to the promoter of miR-34a, a well-known tumor suppressor miRNA, repressing its transcription through deacetylation of H3K18ac [76].

#### 2.2.5. HDAC of Class IV

**HDAC11:** To date, no study has looked at HDAC11 in the context of GC.

#### 2.2.6. Conclusions

Taken together, publications addressing the expression and role of HDACs in GC are statedly increasing but are still far less numerous than in other cancers such as lung or colon cancer. This certainly is the main reason why some HDACs published data do not allow to draw a clear conclusion on their expression pattern or function or show contradictory observations. However, there is quite a consensus in the literature that HDAC4, HDAC7, HDAC9, SIRT2, and SIRT7 are upregulated in GC and seem to be pro-tumor factors, whereas SIRT4, SIRT5, and SIRT6, which are downregulated, seem to have a tumor suppressor function (Table 2). Similarly, although some minor contradictions exist, HDAC1 and HDAC3 have been mostly shown to be overexpressed in GC and associated with a poor prognosis (Table 2). Several publications showed implication of HDACs in survival indicating that some of them seem to be potential prognostic biomarkers as HDAC3, HDAC4, HDAC5, HDAC6, HDAC7, SIRT3, SIRT5, SIRT6, and SIRT7, and among these, HDAC1, HDAC2, and HDAC4 may also be predictive biomarkers (Table 3). However, much more work is still needed to comprehensively understand their role in GC progression before they can be considered as potential biomarkers or therapeutic targets.

## 3. Histone Deacetylase Inhibitors

### 3.1. Generalities

Because HDACs are deregulated in numerous cancers and involved in many biological processes, inhibitors of HDACs (HDACis) seem to be a promising new class of compounds to treat cancer.

Among HDACis, there are compounds of synthetic or natural origin and, as HDACs, they are divided into groups: hydroxamates, aliphatic acids, benzamides, depsipeptides, also referred to as bicyclic peptides, and SIRT inhibitors.

Hydroxamates inhibit class I and II HDACs by binding to their zinc ion in the catalytic domain [20]. Among these, several HDACis have been developed and approved by the FDA to treat different types of cancer. First, Vorinostat (suberoylanilide hydroxamic acid, SAHA) was approved by the FDA in 2006 to treat cutaneous T-cell lymphoma (CTLC) in patients with progressive, persistent, or recurrent disease on or following two systemic therapies [111]. It is known as Zolina^®^, an orally active produced by Merck. Secondly, Belinostat, another hydroxamic acid known as Beleodaq^®^, fabricated by Topotarget and Spectrum Pharmaceuticals, was approved by the FDA in 2014 to treat patients with relapsed or refractory peripheral T-cell lymphoma (PTCL) by intravenous route [112]. Panobinostat was the latest to be approved by the FDA in the context of cancer treatment. It was approved in 2015 and is named Farydak^®^, manufactured by Novartis Pharmaceuticals. It is intended for patients with multiple myeloma who have received two prior regimens, including bortezomib and an immunomodulatory agent [113]. Panobinostat and dexamethasone (immunomodulatory agent) are administrated orally and bortezombib, intravenously.

Aliphatic acids compose another group of HDACis. They have short-chained fatty acid structure. None have been approved by the FDA for cancer treatment. Several aliphatic acids are nevertheless well known and studied, such as valproic acid (VPA), which has been approved for treatment of epilepsy, bipolar disorder, schizophrenia, or migraine [20].

Benzamides, the third group of HDACis, include among their best-known members entinostat (MS-275) and mocetinostat (MGCD0103) [14].

Depsipeptides or bicyclic peptides have a tetrapeptide structure. One HDACi of this group has been approved by the FDA: romidepsin, named Istodax and manufactured by Celgene Corporation. It was approved to be administrated intravenously in patients with CTLC or PTCL, who have received one prior therapy in both cases, in 2009 and 2011, respectively [114].

Although several inhibitors for SIRT are published such as nicotinamide, cambinol, and sirtinol derivatives [20], their diversity does not allow for their unification into specific groups.

To conclude, HDACis can impact many biological functions and can have a high impact on the therapeutic outcomes of cancers, including T-cell lymphoma and myeloma, where four of them have been approved for clinical used by the FDA. Further, many studies have revealed a promising role of HDACis as therapeutic agents in various cancers other than those mentioned above [115]. In addition, this review shows that HDACs are all deregulated in GC (see above, except HDAC11 for which there is no information). In this respect, this review focuses on what is known in GC about FDA-approved HDACis: belinostat, panobinostat, romidepsin, and vorinostat.

### 3.2. In Gastric Cancer

#### 3.2.1. Belinostat

To date, no study has examined **belinostat** in the context of GC.

#### 3.2.2. Panobinostat

**Panobinostat** is still poorly studied in GC with only three publications addressing its effect on GC cell progression and drug potential for the treatment of GC. In general, panobinostat seems to inhibit cell viability and proliferation and to induce the expression of apoptosis-associated proteins, including cleaved PARP and cleaved caspase-3 in different GC cell lines [38,116,117]. In vivo, panobinostat represses tumor progression of the GC cell line in immunodeficient mice [116]. According to Regel and co-workers, panobinostat increases expression of CITED2 to repress anthracycline-resistance gene expression [38]. This leads to improve anthracyclines efficiency to reduce tumor growth in GC. Taken together, these results suggest that panobinostat could be used to treat GC in combination with chemotherapies such as anthracyclines. In this respect, an open-label, phase II trial evaluating the antitumor activity and safety of the oral HDACi LBH589 has been enrolled but, to our knowledge, no outcome of this trial has been published. The treatment consisted in 20 mg LBH589 three times a week in patients with chemo-refractory HDAC overexpression (ClinicalTrials.gov Identifier: NCT01528501).

#### 3.2.3. Romidepsin

The effect of **romidepsin** on GC cells was mostly studied in combination with different therapeutic agents. When combined with adenovirus-mediated p53 family gene therapy [118], ionizing radiation (IR) [119], or with the proteasome inhibitor bortezomib [120], romidepsin increases apoptosis in cells and reduces tumor growth in GC xenograft nude mice. In addition, romidepsin seems to be able to induce EBV-lytic cycle in EBV positive GC cell lines, resulting in increased cell death and reduced tumor growth in mice [121]. Interestingly, a combination of romidepsin with the DNA methyltransferase inhibitor 5-aza-2′-deoxycytidine (5-Aza-CdR) or with oxaliplatin displays an synergistic anti-proliferative effect, whereas the combination of all three drug results in an antagonistic effect [122]. To conclude, all these studies highlight the therapeutic potential of romidepsin when combined with other anti-cancer drugs. There is one open-label, multi-center phase II study published with the primary objective to determine the radiographic response rate (complete response and partial response) in patients with refractory adenocarcinoma of the stomach or gastroesophageal junction with FR901228 (Romidepsin). Patients received FR901228 over 4 h on days 1, 8, and 15, and in absence of disease progression or unacceptable toxicity, courses were repeated every 28 days. Unfortunately, no outcome of this study is published so far (ClinicalTrials.gov Identifier: NCT00098527).

#### 3.2.4. Vorinostat

**Vorinostat** is the most studied pan-HDACi in GC. All studies agreed on its tumor suppressive role in GC. Many papers highlighted that vorinostat inhibits cell proliferation and induces apoptosis, cell cycle arrest, and autophagy in GC cell lines and decreases tumor growth in mice [22,64,123,124,125,126,127,128,129]. The molecular mechanisms underlying these effects seem to include reactivation of RunX3, a tumor suppressor (127), and inhibition of the enhancer zeste homolog 2 (EZH2) expression, which subsequently leads to the activation of the suppressive miRNAs, miR-1246 and -302a [126] and an induction of miR-769-5p/-3p targeting the pro-tumoral complex STAT3/IGF1R/HDAC3 [130]. Furthermore, expression of the B-cell translocation gene (BTG) family members, *BTG1* and *BTG3*, seems to enhance the sensitivity of GC cells to vorinostat, suggesting that they can be used as targets for gene therapy when combined with vorinostat [131,132]. In contrast, expression of c-MYC, regulating the expression of MCL1 and eIF4E/BCL_XL_ [133]; ribonuclease inhibitor (RNH1), impacting on the cellular redox status [134]; and inhibitor of growth protein 5 (ING5), regulating β-catenin and NF-κB and the Akt pathway [135], seems to mediate resistance to vorinostat, suggesting that their inhibition in combination with vorinostat could be a therapeutic solution for GC patients. Interestingly, some studies showed that, depending on the particular molecular subtype of the GC, vorinostat might be more efficient. Similar to romidepsin, vorinostat reactivates the EBV lytic cycle in EBV-positive GC cells to improve their death compared to EBV-negative GC cells suggesting that, in addition to romidepsin, another HDACis similar to vorinostat could be a new therapeutic strategy to treat EBV-positive GC patients [136]. However, it is also shown that vorinostat reduces cell proliferation and increases apoptosis in CDH1-deficient cells, suggesting that they are more sensitive than WT cells, and that vorinostat could be used in the treatment of hereditary diffuse GC, whose mutated *CDH1* is a hallmark [125]. Unfortunately, in the case of *H. pylori*-associated GC, vorinostat induces the expression of CAPZA1 (capping actin protein of muscle Z-line alpha subunit 1), resulting in an escape of CagA from autophagic degradation [137]. More precisely, Tsugawa and co-workers showed that vorinostat treatment of AGS cells induces the acetylation of histone H3 at the proximal *CAPZA1* promoter region, inducing its expression. CAPZA1 inhibits LAMP1 (lysosomal associated membrane protein 1) via binding to LRP1-ICD (LDL receptor related protein 1 intracellular domain), resulting in inhibition of autolysosomal formation [137].

Interesting, vorinostat seems to have a profound impact on the tumor micro-environment. Deng and co-workers demonstrated that vorinostat inhibits the expression of B7-H1, leading to an increased percentage of tumor-infiltrating CD8+ T cells in GC mouse models [22]. In another murine model, GC patient-derived xenograft (PDX) mice, Venkatasamy et al. showed that vorinostat reduces cell proliferation and growth, modifies the internal tumor structure, and favors a mesenchymal-cell-like gene signature [138].

Some studies investigated the effect of combinatory treatment of vorinostat with either cytotoxic drugs or drugs impacting on cellular homeostasis, cell survival, and tumor growth. Several studies were interested in the combination of vorinostat with chemotherapies. Vorinostat in combination with taxanes, such as paclitaxel and docetaxel [123], cisplatin [33], and oxaliplatin [129], is more efficient in inhibiting tumor growth than the drug alone in GC cells and mice. According to Zhou et al., vorinostat improves oxaliplatin effects by inhibiting oxaliplatin-induced Src activation through its phosphorylation [129]. Seah et al. showed that vorinostat in combination with doxorubicin and/or cisplatin induces DNA damages and apoptosis, highlighting a synergistic effect with doxorubicin whereas an additive effect with cisplatin [139]. Vorinostat has also been shown to induce an autophagy gene expression signature in GC cells and bafilomycin A1, an autophagy inhibitor, increases its anti-proliferative effect [124]. Similarly, the proteasome inhibitor MG132 displays a synergistic effect with vorinostat to inhibit proliferation and glycolysis and to induce cell cycle arrest and apoptosis in GC cell lines [128].

However, the above study also pointed out to some potential side effects that vorinostat might have when used for the treatment of GC, as at low concentrations it increases cell migration and invasion, and its combination with MG132 damages hepatic functions [128].

Phase I and II clinical trials of vorinostat in combination with capecitabine plus cisplatin as a first-line chemotherapy for GC patients with unresectable or metastatic cancer were already performed [140,141]. Patients were treated with 400 mg vorinostat once daily on days 1–14, 1000 mg ^m−2^ capecitabine twice daily on days 1–14, and 60 mg ^m−2^ cisplatin on day 1 every 3 weeks. They showed that objective response reached 42% and median PFS and overall survival was 5.9 and 12.7 months, respectively [141]. These results and the increase in side effects led to the conclusion that the combined treatment does not improve outcomes of GC patients compared to standard fluoropyrimidine–platinum doublet regimens in GC patients. However, note that the patients were not stratified in any way, neither on potential overexpression of HDACs nor on any other potential mechanisms that might be important for HDACi activity. In addition, side effects and/or lack of efficacy might also be due to the unspecific activity of the vorinostat, which affects several HDACs, and its action on the cancer cells as well as the microenvironment. Using HDACi selective of a given HDAC or using vectorized compounds, that may target selectively the cancer cells, might be an interesting strategy to pursue.

To conclude, vorinostat seems to have an anti-tumoral function and to improve the effectiveness of chemotherapy in vitro and in vivo, but presents some toxicities in clinical trials, showing that further investigations in combination with different drugs will be necessary to use it for GC patients.

#### 3.2.5. Conclusions

In conclusion, panobinostat, romidepsin, and vorinostat act as tumor suppressors in GC. These HDACis inhibit cell proliferation and induce apoptosis in GC cells and suppress tumor growth in GC mouse models. Different studies showed that some are also implicated in cell cycle arrest, and inhibition of migration, metastases formation, and tumor suppressor gene expression. Many pathways and factors seem to be involved in their function as, for example, tumor suppressors [127], miRNA [126,130], or BTG family members [131,132].

In addition, vorinostat and romidepsin show potential therapeutic value in treating EBV-positive GC patients [121,136] and patients with initial hereditary diffuse GC [125]. Additionally, according to several studies in GC cells and mice, combinatory treatments with HDACis and different therapies could be beneficial for the treatment of GC, such as chemotherapies, [33,38,123,129,139], radiotherapy [119], and gene therapy [118], or other agents such as DNMTi [122], autophagy inhibitors [124], and proteasome inhibitors [120,128].

Despite the promising effects of HDACis in GC, their study will have to be deepened, concerning vorinostat in particular, to overcome the clinical side effects observed and to obtain better results than the current treatments.

## Figures and Tables

**Figure 1 cancers-14-05472-f001:**
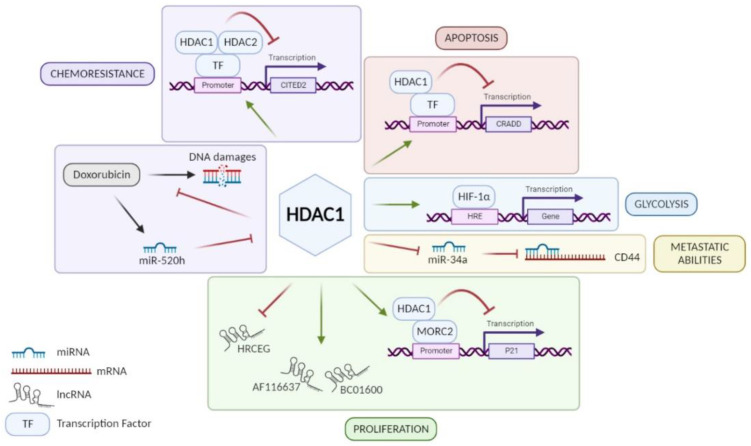
Role of HDAC1 in different signaling pathways involved in GC progression. HDAC1 promotes GC cell proliferation via lncRNA HRCEG repression, lncRNAs BC01600 and AF116637 upregulation, or MORC2/p21 pathway. HDAC1 favors metastatic abilities of GC cells through miR-34a repression inducing CD44 expression. HDAC1 promotes HIF-1α activity leading to glycolysis in GC cells. HDAC1 inhibits CRADD transcription and consequently caspase-2-dependent apoptosis. HDAC1 is a resistance factor to chemotherapies in GC such as anthracycline, via repression of CITED2, and doxorubicin, inhibiting its fixation on DNA. Figure has been generated using Biorender (Biorender.com).

**Table 1 cancers-14-05472-t001:** Classification of HDACs and their subcellular localization.

Family	Class		Member	Yeast Counterpart	Subcellular Localization
Zn^2+^-dependent	I		HDAC 1, 2, 3, 8	Rpd3	Nucleus
	II	a	HDAC 4, 5, 7, 9	Hda1	Nucleus/Cytoplasm
		b	HDAC 6, 10		
	IV		HDAC 11		Nucleus
NAD^+^-dependent	III		SIRT 1, 2	Sir2	Nucleus/Cytoplasm
			SIRT 3, 4, 5		Mitochondria
			SIRT 6, 7		Nucleus

**Table 2 cancers-14-05472-t002:** Summary of HDAC regulations, targets, cellular process, and clinicopathological implications in GC. *Disputed.

Class	Member	Regulation	Target	Cellular Process Implication	Clinicopathological Implication	Reference
**I**	**HDAC1**	Up, None, or Down*	CITED2 CRADD HIF-1α lncRNA AF116637 lncRNA BC01600 lncRNA HRCEG mir-34a/CD44 p21	Cell proliferation Apoptosis Glycolysis EMT Chemosensitivity	Age *H. pylori* infection Lauren classification Lymph node metastasis Lymphovascular invasion Tumor size Tumor stage Survival*	[21,23,24,25,26,27,28,30,32,33,34,35,36,37,38,39,40,41]
	**HDAC2**	Up or None*	CITED2 p16^INK4a^	Cell proliferation Autophagy Apoptosis Chemosensitivity	Lauren classification* Tumor grade Tumor stage Survival*	[29,33,38,42,44,44,45,46,48]
	**HDAC3**	Up or None*	FOXA2/FTO/MYC miR-376/WNT2b mir-454/CHD5	Cell proliferation Apoptosis Cell invasion Cell migration Tumor growth	LNM Tumor grade Tumor infiltration depth Tumor stage Survival	[22,31,47,49,50,51,52]
	**HDAC8**	Up or None*		Cell cycle arrest Cell proliferation Apoptosis	LNM Tumor grade Tumor stage	[47,53]
**IIa**	**HDAC4**	Up	ATG4B/MEKK3/p38 BIK MIAT/miR-29a-3p P21	Cell cycle arrest Cell proliferation Apoptosis Autophagy Cell invasion Cell migration Tumor growth Chemosensitivity	*H. pylori* infection Lauren’s classification LNM Molecular subgroup Tumor depth invasion Tumor stage Survival	[32,54,55,56,57,58]
	**HDAC5**	Up or Down*	HOXC-AS3/YBX1	Cell proliferation Cell migration	Survival	[43,59,60,61]
	**HDAC7**	Up		Cell proliferation Cell migration Cell invasion	Survival	[63]
	**HDAC9**	Up or None*		Cell proliferation Apoptosis Tumor growth	Tumor stage* Survival*	[64,65,66]
**IIb**	**HDAC6**	Up or Down*	EGFR/Rabaptin-5	Cell proliferation	H. pylori infection Tumor progression Survival	[32,67]
	**HDAC10**	Down				[68]
**III**	**SIRT1**	Up or Down*	AMPK/FOXO3c-JUN/ARHGAP5 miR-12129 miR-132/SIRT1/CREB/ABCG2 miR-204/SIRT1/LKB1 NFκB/cyclin D1 STAT3/MMP-13 Yap/SIRT1/Mfn2	Cell proliferation Apoptosis Autophagy Senescence Cell invasion Cell migration EMT Mitophagy Chemo-sensitivity*	Sex Age Lauren’s classification Tumor grade Tumor size Tumor invasion LNM Tumor stage Survival*	[69,70,71,72,73,74,75,76,77,78,79,80,81,82,83,84,85,86,87,88,89,90,91,92,93]
**III**	**SIRT2**	Up	PEPCK1	Cell cycle arrest Cell proliferation Apoptosis Cell invasion Cell migration Tumor growth Metastases	Survival	[94,95]
	**SIRT3**	Up or Down*	HIF-1α LDHA Notch-1	Cell proliferation Cell invasion ROS homeostasis Tumor growth	Tumor grade Surival	[96,97,98,99,100,101,102]
	**SIRT4**	Down	E-cadherin	Cell proliferation Cell invasion Cell migration EMT	LNM Tumor stage Survival	[103,104,105]
	**SIRT5**	Down	OGDH S100A10	Cell cycle arrest Cell proliferation Cell invasion Cell migration EMT	Lymphovascular invasion Nodal involvement Survival	[106,107,108,109]
	**SIRT6**	Down	JAK2/STAT3	Cell cycle arrest Cell proliferation Apoptosis Tumor growth	Tumor grade Tumor size Tumor stage Survival	[110]
	**SIRT7**	Up	mir-34a	Cell proliferation Apoptosis Tumor growth	Extent of gastrectomy Lymph node status Tumor depth invasion Tumor size Metastases Tumor stage Survival	[76]
**IV**	**HDAC11**	no data available				

**Table 3 cancers-14-05472-t003:** Characterization of HDACs as potential diagnostic, prognostic, or predictive biomarkers. Green: potential biomarker; red: controversial biomarker; white: no indications.

Class	Member			
		Diagnostic	Prognostic	Predictive
**I**	**HDAC1**			
	**HDAC2**			
	**HDAC3**			
	**HDAC8**			
**IIa**	**HDAC4**			
	**HDAC5**			
	**HDAC7**			
	**HDAC9**			
**IIb**	**HDAC6**			
	**HDAC10**			
**III**	**SIRT1**			
	**SIRT2**			
	**SIRT3**			
	**SIRT4**			
	**SIRT5**			
	**SIRT6**			
	**SIRT7**			
**IV**	**HDAC11**	no	no	no

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
