# Peer review of "Histone Deacetylase Functions in Gastric Cancer: Therapeutic Target?"

_cancers, 2022, doi:10.3390/cancers14215472_

Round 1

Reviewer 1 Report

The present review from Badie et al. is an interesting and well-written summary of the role of histone deacetylases in Gastric Cancer as well as their potential use as molecular targets for the treatment of these tumors.

Only a few minor issues should be addressed. Consider the following points to further improve the review:

-       please review lines 114-117, they do not seem to make sense.

-       In the current version of the manuscript, the tables are acceptable although sometimes there is some discordance with what the text describes (for example HDAC4 is related to drug resistance - line 252 - but not HDAC5, contrary to what the tables reflect). Please review the content of the tables for better agreement with the text.

-       Acronyms “LNM” should be described in Table 2

-       Information concerning HDAC11 (or lack of information in the literature) should be indicated differently in table 2 to avoid having an empty row.

-       I suggest further expanding the section describing histone deacetylase inhibitors, providing more information about the efficacy of the drugs, and other data provided by clinical trials.

-       In the text there is a reference to figure 1 that is not available in the manuscript.

Author Response

Response: We thank the reviewer for his constructive comments:

The present review from Badie et al. is an interesting and well-written summary of the role of histone deacetylases in Gastric Cancer as well as their potential use as molecular targets for the treatment of these tumors.

Only a few minor issues should be addressed. Consider the following points to further improve the review:

1) Please review lines 114-117, they do not seem to make sense.
Response: Has been corrected.

2) In the current version of the manuscript, the tables are acceptable although sometimes there is some discordance with what the text describes (for example HDAC4 is related to drug resistance - line 252 - but not HDAC5, contrary to what the tables reflect). Please review the content of the tables for better agreement with the text. We apologize for this discrepancy.

Response: True current available data don't link HDAC 5 with resistance to treatment and we corrected the text and table accordingly.

3) Acronyms “LNM” should be described in Table 2:
Response: Has been corrected.

4) Information concerning HDAC11 (or lack of information in the literature) should be indicated differently in table 2 to avoid having an empty row.
Response: Has been corrected.

5) I suggest further expanding the section describing histone deacetylase inhibitors, providing more information about the efficacy of the drugs, and other data provided by clinical trials.

Response: We have added the information concerning the treatment protocols used in the different clinical trials.

In this respect an open-label, phase II trial evaluating the antitumor activity and safety of the oral Histone Deacetylase (HDAC)-Inhibitor LBH589 has been enrolled but, to our knowledge, no outcome of this trial has been published. The treatment consists of 20 mg LBH589 three times a week in patients with chemo-refractory HDAC overexpres-sion (ClinicalTrials.gov Identifier: NCT01528501).

Patient were treated with 400mg vorinostat once daily on days 1-14, 1000mg m-2 capecitabine twice daily on days 1-14, and 60 mg m-2 cisplatin on day 1 every 3 weeks.

There is one open-label, multicenter phase II study published with the primary objective to determine the radiographic response rate (complete response and partial response) in patients with refractory adenocarcinoma of the stomach or gastroesophageal junction with FR901228 (Romidepsin). Patients received FR901228 over 4 hours on days 1, 8, and 15, and in absence of disease progression or unacceptable toxicity courses were repeated every 28 days. Unfortunately, no outcome of this study is published so far (ClinicalTrials.gov Identifier: NCT00098527).

6) In the text there is a reference to figure 1 that is not available in the manuscript.
Response: As an arrow of submission Figure 1 has been submitted as "graphical abstract" and was maybe therefor not included in the manuscript. This should be corrected in new version.

Reviewer 2 Report

The authors have well described the functional role of HDAC and SIRT in gastric cancer.

  I expect that this paper will be an excellent reference for researchers studying gastric cancer in the future.

However, there are some things I would like to add.

The most common cause of gastric cancer is infection with Helicobacter pylori.

I am sure that the completeness of the paper will be improved if the function of HDAC by Helicobacter pylori infection is explained in more detail.

The acetylation by Helicobacter pylori infection, which is involved in the development of gastric cancer, is also an area of interest for many researchers.

Also, I request that the content related to MKN28 be deleted from the manuscript as a minor correction. Since MKN28 is a cell with a contamination issue, we do not think it is suitable for gastric cancer review papers.

Author Response

We thanks the reviewer for his constructive evaluation of our manuscript, and the following changes have been made:

1) The most common cause of gastric cancer is infection with Helicobacter pylori. I am sure that the completeness of the paper will be improved if the function of HDAC by Helicobacter pylori infection is explained in more detail.

Response: The acetylation by Helicobacter pylori infection, which is involved in the development of gastric cancer, is also an area of interest for many researchers. We do agree that infection of Helicobacter pylori is an important issue in gastric cancer development. We have extended parts in the manuscript discussing in more detail the relationship between H. pylori and certain HDACs and added the respective reverences.

The following information has been added to the manuscript:

HDAC1: According to some studies, high expression of HDAC1 in GC can be associated with age, Lauren’s classification, H. pylori infection, tumor size, lymphovascular invasion, lymph node metastasis (LNM), or even advanced tumor stages (Gao et al., 2012; He et al., 2017; Jiang et al., 2017, 2019; Mutze et al., 2010; Sudo et al., 2011; Sun et al., 2020).

HDAC1: For example, HDAC1 promotes cell proliferation via the HDAC1/MORC2/p21 pathway (Zhang et al., 2015a). However, H. pylori infection seems to reverse the repressive activity of HDAC1 on p21 promoter in GC cells and thus promotes acetylation of the latter, leading to p21 expression (Xia et al., 2008). These results agree with observations made by Pero et al. which showed that H. pylori decreases expression and activity of several HDACs such as HDAC1 (Pero et al., 2011).

HDAC6: He and co-workers showed that HDAC6 expression progressively reduces during progression from precancerous conditions to GC (He et al., 2017). Besides, the authors showed that HDAC6 is inversely correlated with H. pylori infection in gastric mucosal lesions (He et al., 2017). In vitro and in vivo, infection of GES-1 cell line or mice by SS1 strains decreased HDAC6 expression suggesting that H. pylori can be involved in its downregulation in GC (He et al., 2017).

Vorinostat: Unfortunately, in the case of H. pylori associated GC vorinostat (SAHA) induces the ex-pression of CAPZA1 (capping actin protein of muscle Z-line alpha subunit 1) resulting in an escape of CagA from autophagic degratdation (Hitoshi et. al., 2019). More precise-ly, Hitoshi and co-workers have shown that SAHA treatment of AGS cells induces the acetylation of histone H3 at the proximal CAPZA1 promoter region induces its expres-sion. CAPZA1 inhibits than LAMP1 (lysosomal associated membrane protein 1) via binding to LRP1-ICD (LDL receptor related protein 1 intracellular domain) resulting in inhibition of autolysosomal formation (Hitoshi et. al., 2019).

2) Also, I request that the content related to MKN28 be deleted from the manuscript as a minor correction. Since MKN28 is a cell with a contamination issue, we do not think it is suitable for gastric cancer review papers.

Response: The information concerning MKN28 been deleted from the text.

Reviewer 3 Report

Badie et. al., current review is based on the most aggressive cancer which is gastric cancer. In the present study the author have summarized the various HDAC inhibitors as the potential chemotherapeutic agents. They have also discussed in detail about the 18 HDACs in gastric cancer including their cellular processes, their criminal importance and impacts on patients. In conclusion, the authors identified panobinostat, romidepsin and vorinostat as the potential tumor suppressants. Despite their side effects the HDAC’s still remain to the most promising targets for GCs.

Minor comments:

In abstract, line 18, HDACis should be corrected to HDAC’s

The author is recommended to undergo a thorough check of the manuscript for typographical and grammatical errors.

Author Response

We thank the reviewer for his positive comments.

Minor comments:

1) In abstract, line 18, HDACis should be corrected to HDAC’s.

Response: Line 18 revers to the FDR approval of HDAC inhibitors (HDACi). We have changed HDACis to HDACi's.

2) The author is recommended to undergo a thorough check of the manuscript for typographical and grammatical errors.

Response: Has been done.